# Perturbation of Circadian Rhythm Is Associated with Increased Prevalence of Chronic Kidney Disease: Results of the Korean Nationwide Population-Based Survey

**DOI:** 10.3390/ijerph19095732

**Published:** 2022-05-08

**Authors:** Yina Fang, Serhim Son, Jihyun Yang, Sewon Oh, Sang-Kyung Jo, Wonyong Cho, Myung-Gyu Kim

**Affiliations:** 1Department of Internal Medicine, Korea University Anam Hospital, Seoul 02841, Korea; fangyn1994@gmail.com (Y.F.); coo-kie@daum.net (J.Y.); sewon_oh@korea.ac.kr (S.O.); sang-kyung@korea.ac.kr (S.-K.J.); wonyong@korea.ac.kr (W.C.); 2Department of Biostatistics, Korea University College of Medicine, Seoul 02842, Korea; sonserhim@korea.ac.kr

**Keywords:** chronic kidney disease, sleep onset time, sleep duration, shift work

## Abstract

Disturbances in circadian rhythms cause several health problems, such as psychosis, metabolic syndrome, and cancer; however, their effect on kidney disease remains unclear. This study aimed to evaluate the association between chronic kidney disease (CKD) and sleep disturbance in a Korean adult population. A total of 17,408 participants who completed the National Health and Nutrition Examination Survey from 2016 to 2018 were assessed for their sleep patterns and renal function. CKD was defined as an estimated glomerular filtration rate ≤ 60 mL/min/1.73 m² or a positive dipstick urinalysis. Sleep onset time and sleep duration showed significant differences between the control and CKD groups (*p* < 0.001). After adjusting for the covariates, sleep onset time rather than sleep duration was independently associated with incidence of CKD, and this association was more significant in people who were older, in women, and in those with low body mass index and no comorbidities. When comparing the prevalence of newly diagnosed CKD according to sleep onset time in a population with no CKD risk factors or no history of CKD, the early bedtime group showed an independent association with incidence of new CKD (odds ratio (OR), 1.535; 95% confidence interval (CI), 1.011–2.330) even after adjusting for covariates. Impaired circadian rhythm along with sleep disturbance could be associated with CKD development; therefore, sleep disturbance might be an important therapeutic target for CKD.

## 1. Introduction

Different physiological functions from all levels of behavioral response and molecular interaction adapt to changes in the environment according to the cycle of day and night and exhibit circadian rhythms. In humans, these circadian rhythms are coordinated by molecular clock work of the brain and peripheral organs, which are synchronized with optical signals or other zeitgebers to respond to daily environmental changes [1]. The coordination of circadian rhythms plays an important role in maintaining physiological homeostasis, such as cell cycle progression, cytokine release, sleep–wake cycle, immune activity, renal function, and hepatic glucose metabolism [2,3]. Given the importance of the circadian system in human physiology, disruption of circadian rhythm caused by environmental factors such as changes in sleep onset, exposure to night lights, shift work, or jet lag can lead to a number of health problems, including obesity, psychiatric disorders, cancer, and metabolic and cardiovascular diseases [4,5].

The kidneys are organs optimized for circadian rhythms, and are particularly rich in clock genes that serve as an important regulator of various renal functions including circadian changes in renal plasma flow, tubular transporter expression, hormone secretion, and glomerular filtration rate [6,7,8,9]. Thus, environmental factors that disturb the circadian rhythm can potentially disturb the rhythmic oscillations of clock genes and renal function. For example, acute sleep deprivation in healthy adults was reported to cause nocturnal polyuria and attenuate nocturnal blood pressure dipping [10]. Animals with disrupted molecular clocks also exhibited abnormal blood pressure and impaired circadian rhythms of water and sodium homeostasis [9]. The circadian rhythm is not only a regulator of kidney function but also appears to have important implications for the treatment of hypertension and kidney disease. A few animal studies have reported a close association between chronic kidney disease (CKD) and the circadian clock system. Motohashi et al. [11] demonstrated that adenine-induced kidney injury was exacerbated in clock gene mutant mice and it was accompanied with non-dipping hypertension and lower heart rate, suggesting that disruption of the circadian clock can contribute to CKD progression. In humans, several cohort studies have reported that poor sleep quality and lack of sleep are associated with CKD development [12,13]. Although they suggest a possible effect of asynchronous circadian rhythm on kidney disease, poor sleep duration or sleep quality does not necessarily indicate a disturbance in the circadian rhythm.

Given that the light-dark cycle has a strong effect on the circadian cycle, sleep onset time may be more important than sleep quality to maintain a physiological circadian rhythm. Therefore, to elucidate the role of circadian rhythm on CKD, we investigated the relationship between sleep onset time and CKD development, and also compared the effects of work schedule and sleep duration on CKD based on a nationwide population-based survey.

## 2. Materials and Methods

### 2.1. Study Design and Population

This study used data from the Korea National Health and Nutrition Examination Survey (KNHANES) from 2016 to 2018. KNHANES is a nationwide stratified multi-level complex database for collecting representative samples of the Korean population using city, town, and house types as strata and additionally sex, age, and residential area ratio as implicit stratification variables. Data collection was conducted annually by the Korea Centers for Disease Control and Prevention [14]. Out of the 24,389 survey participants, we included 19,389 adults over 19 years old. Among them, 1061 participants were excluded due to insufficient laboratory data. Additionally, participants who did not provide information on their sleep parameters were also excluded (*n* = 920). Finally, a total of 17,408 participants met the inclusion criteria, including 963 patients with CKD (Figure 1). In addition, the analysis of work schedule was restricted to participants who responded to questions about work schedules. A total of 11,556 participants were enrolled in the analysis, including 431 subjects with CKD. CKD was defined by an estimated glomerular filtration rate (eGFR) ≤60 mL/min/1.73 m², calculated using the Chronic Kidney Disease Epidemiology Collaboration equation, or a positive dipstick urinalysis. Written informed consent was obtained from each participant in the KNHANES at the time of enrollment. The study protocol for the survey was approved by the institutional review board of the Korea Centers for Disease Control and Prevention (IRB No. 2018-01-03-P-A).

### 2.2. Assessment of Sleep Parameters

The sleep parameters included total sleep duration and sleep onset time. Sleep duration was assessed by the question “how many average hours of sleep do you get a day?” The responses were then categorized into <6 h, 6–8 h, and >8 h. In addition, renal function and CKD prevalence were compared between 2 h intervals of bedtime or between early (18:00–22:00), mid (22:00–02:00), and late (02:00–6:00) sleep onset.

### 2.3. Assessment of Work Schedule

Work schedule was categorized as: day shift (6:00–18:00), afternoon shift (14:00–24:00), night shift (21:00–8:00), split shift work (work split into two or more parts), and 24 h rotating shift.

### 2.4. Assessment of Covariates

The covariates included sociodemographic variables (age, sex, and education level), body mass index (BMI), smoking status, drinking status, physical activity, comorbidities (hypertension, diabetes, hyperlipidemia, cerebrovascular diseases, myocardial infarction, and angina pectoris), systolic blood pressure (SBP), diastolic blood pressure (DBP), creatinine levels (Cr), fasting blood glucose levels (FBG), and triglyceride levels (TG).

### 2.5. Statistical Analyses

Statistical analyses were performed using SAS version 9.4 (SAS Institute, Inc., Cary, NC, USA). Differences in the baseline characteristics between the patients with CKD and the controls were analyzed using the chi-squared test or Student’s *t*-test. The association between sleep, work schedule, and CKD was assessed using multivariable logistic regression analysis, whereby covariates from four progressively adjusted models were as follows: Model 1 was a crude model without adjustment; Model 2 included both age and sex; Model 3 included age, sex, BMI (<25 kg/m^2^, ≥25 kg/m^2^), educational level (completion of middle school or higher than high school), smoking status (no or yes (current or ex-smokers)), drinking status (no or yes (drinking alcohol at least once in the past 12 months)), physical activity(≥2 days/week, <2 days/week), SBP, DBP, GLU, TG, and comorbidities (such as diabetes, hypertension, hyperlipidemia, angina, myocardial infarction, and stroke); Model 4 included age, sex, education level, smoking, drinking, physical activity, and BMI; Model 5 included age, sex and work schedule (day shift, afternoon shift, night shift, split shift, and 24 h rotating shift); and Model 6 included age, sex, education level, smoking, drinking, physical activity, BMI, work schedule, and sleep duration (<6 h/day, 6–<8 h/day, ≥8 h/day). Significance between each CKD incidence by sleep onset time was calculated by Pearson’s chi-square test. Statistical significance was defined as *p* < 0.05.

## 3. Results

### 3.1. General Characteristics of the Participants

The general demographic and clinical characteristics of the participants are shown in Table 1. A total of 963 of the 17,408 participants belonged to the CKD group. Participants in the CKD group were more likely to be older and smoke, had a higher prevalence of diabetes and hypertension, and had higher BMI and triglyceride levels. Moreover, they were also less physically active and less educated. Although there have been recent studies on the relationship between low educational attainment and CKD [15], our data could not explain the causal relationship, and no additional analysis of the factors supporting the association was performed. In addition, there was a significant difference between the two groups on age, which may affect statistical differences in several variables. To alleviate these concerns about the effects of aging, the difference between the two groups was reanalyzed by applying the definition of CKD including an age-specific threshold for GFR proposed by a study group [16], but similar statistical significance was observed (Appendix A).

### 3.2. Association between Sleep Disturbance and CKD

Sleep onset time and sleep duration showed significant differences between the control and CKD groups (*p* < 0.001, Table 1). In the univariable analysis, early bedtime was associated with a higher CKD incidence (odds ratio (OR): 2.599, 95% confidence interval (CI): 2.228–3.031). However, no independent association was found in the models adjusted for age, sex, smoking status, alcohol consumption, education level, activity level, and comorbidities (hypertension, diabetes, hyperlipidemia, cerebrovascular diseases, myocardial infarction, and angina pectoris). In contrast, the late bedtime group showed an independent association with CKD incidence in Models 2 and 3, which were adjusted for various risk factors (OR: 1.531, 95% CI: 1.135–2.066 and OR: 1.479, 95% CI: 1.083–2.020, respectively, Table 2). When comparisons of the early, mid, and late sleep onset groups were stratified by age, sex, BMI, education level, smoking, drinking and comorbidities, the association between late bedtime and CKD was stronger in older adults, women, and people with lower BMI (*p* < 0.05). However, no significant interactions were found in each stratum (Appendix A).

In the sleep duration analysis, the group with long sleep durations (≥8 h/day) had a higher incidence of CKD than the group with normal (6 to <8 h/day) or short sleep duration (<6 h/day), but the analysis, adjusted for various risk factors, showed marginal significance only when compared with the group with short sleep durations (Table 2).

The significant association between sleep onset time and CKD cannot exclude the possibility that more frequent sleep disturbances were observed in pre-existing CKD patients. Therefore, to clarify the direct effect of sleep onset time on CKD development, we further analyzed the prevalence of newly diagnosed CKD in a healthy population with no history of CKD and no risk factors for CKD such as diabetes, hypertension, hyperlipidemia, and cardiovascular disease (Table 3 and Appendix A). In the analysis, we observed that the prevalence of CKD was significantly higher in the early bedtime group even after adjusting for confounding factors (OR: 1.535, 95% CI: 1.011–2.330), suggesting that circadian rhythm disturbances may have acted as non-classical CKD risk factors. When we further subdivided sleep onset time to compare the prevalence of CKD, we found that the prevalence of CKD was U-shaped with bedtime and was the highest for an early bedtime in the population with or without CKD risk factors (Figure 2).

### 3.3. Work Schedule and CKD

Shift work is widely recognized as a reason for inadequate sleep and abnormal bedtimes. To investigate whether work schedule played an important role in the close association between sleep onset and CKD, we first evaluated the relationship between eGFR levels and work schedule. A total of 11,556 participants provided information on the work schedule. Compared with the day work group, significantly decreased eGFR levels were observed in the 24 h rotating shift work group, regardless of CKD risk factors (Figure 3). However, when albuminuria was added as a definition of CKD, work schedule was no longer significantly associated with CKD development. Additionally, the association between sleep onset and CKD in work schedule responders disappeared after work schedule adjustment (Appendix A).

## 4. Discussion

Patients with advanced CKD or patients undergoing dialysis have a higher prevalence of sleep disorders, including difficulty falling asleep, reversal of sleep patterns, and increased sleep latency [13,17,18,19]. Emotional factors, such as depression, anxiety, and stress, and metabolic factors, such as uremia and anemia, electrolyte imbalance, and dialysis-related factors, can contribute to poor sleep quality. In addition, a recent study showed that, even in the general population, insufficient sleep and poor sleep quality might influence the onset and exacerbation of kidney disease, suggesting a bidirectional relationship between sleep disturbance and kidney disease [13]. However, data regarding the association between CKD and sleep duration are inconsistent. For example, a large prospective cohort study found that a shorter sleep duration was associated with a more rapid decline in renal function [20]. In contrast, a rural community-based prospective cohort study found that women with longer sleep durations (≥9 h/day) had higher serum creatinine concentrations and CKD prevalence [21]. Bo et al. [13] also showed that both short and long sleep durations and poor sleep quality were associated with CKD development. Although there is a growing body of evidence that sleep disturbances affect the development of kidney disease, the underlying mechanisms are still poorly understood. Inflammation or sympathetic activation is a possible mechanism that may be triggered by sleep disturbances. However, recent animal studies suggest that an impaired circadian system may contribute to worsening renal function and that circadian rhythm disturbance is a possible pathway link between sleep disturbance and CKD. Considering that day and night, meal time, and artificial light are important external factors regulating circadian rhythm, sleep onset time, rather than sleep duration, might be an important regulator that maintains the circadian system, but the relationship between sleep onset time and disease has received little attention in clinical studies compared to sleep duration or quality. Recently, S Nikbakhtian et al. [22] analyzed the relationship between sleep onset time and cardiovascular diseases (CVD) in 103,712 UK Biobank participants and found a U-shaped relationship between CVD incidence and sleep onset time, with a higher rate of incidence for participants with early (<10 pm) or late (>11 pm) sleep onset time. Similarly, our study also showed that sleep onset time, not sleep duration, was independently associated with CKD incidence, even after adjusting for the major risk factors for the development of CKD such as diabetes, hypertension, hyperlipidemia, angina, myocardial infarction, and stroke. In particular, CKD incidence in the overall population cannot exclude the possibility of the effect of sleep disturbances that accompany patients with preexisting CKD. Therefore, analysis of the incidence of new CKD in a healthy population with no risk of CKD, no related drug use, and no history of CKD could explain the relationship with sleep onset time more clearly. In this analysis, the incidence of CKD was found to be significantly higher in the early bedtime group in a healthy population. This suggests that sleep onset time may act as a risk factor for the development of CKD. The underlying mechanism linking CKD and sleep onset time may be related to impaired circadian rhythm. People who have problems with early sleep onset are more likely to have early morning wake times, and this group is also defined as having “advanced sleep-wake phase disorder”, one of the circadian rhythm disorders [23,24]. The kidney is rich in genes regulating circadian rhythm, and our biological clocks can control various renal physiological processes, including urinary electrolyte excretion and GFR. Therefore, disturbances in the circadian system due to environmental factors, such as sleep disturbance, may significantly affect renal function. Animal studies have shown that circadian rhythm disruption may induce renal dysfunction. For example, Motohashi et al. [11] found that adenine-induced clock mutant mice had an increased severity of CKD, such as higher matrix metalloproteinase-2 expression in the kidney. In addition, mice with kidney cell-specific knockout of Bmal1 exhibited glomerular hyperfiltration, with changes in the circadian rhythm of urinary sodium excretion [25]. Therefore, in our study, inadequate sleep may disrupt the circadian rhythm from a normal light-dark cycle and the peripheral clock system, which may be associated with the development of CKD.

Inadequate melatonin levels might also affect the association between sleep onset time and CKD. A growing number of epidemiologic studies have indicated that night shift workers have lower levels of urinary melatonin production during the night than regular workers, which may be related to chronotype and exposure to light at night [26]. Melatonin is a circadian rhythm modulator, and its production is inhibited by light. In general, melatonin secretion peaks in the middle of the night between 02:00 and 03:00 [27]. In addition, it plays a pivotal role in human biological processes, such as anti-apoptosis, anti-inflammation, and amelioration of the intrarenal renin–angiotensin system through its antioxidative effect [28,29].

We found that the association between sleep disturbance and CKD was more pronounced in women than in men. These results are similar to those from previous studies that reported that shift-related health concerns were more common in women [30,31]. Women’s social status in daytime domestic work and childcare may increase their vulnerability to night shift work. In addition, biological differences between the sexes can also affect circadian rhythms. In general, women have shorter intrinsic circadian cycles (cycle length) and faster timing of melatonin rhythms than men [32,33]. Although the exact mechanisms mediating health problems caused by circadian rhythm disturbances in women are unclear, the social, cognitive, and physiological differences between men and women may contribute to the development of circadian rhythm disorders in women.

Shift work is a factor contributing to sleep disturbance, and previous studies have shown that shift work plays a hazardous role in renal dysfunction. Zhang et al. [34] reported an association between night shift work and decreased eGFR. In addition, electronic work history data from the United States also showed that night shift work is associated with decreased kidney function [35]. In this study, we also found that the 24 h rotating shift work and split shift work group had lower eGFR levels in a population with no CKD risks. However, this relationship did not appear when extending the definition of CKD. Because these findings cannot clearly explain the interaction between sleep onset, work schedule, and CKD development, large-scale cohort studies with detailed work schedule information are needed.

This study had several limitations. First, due to the cross-sectional design of this study, although the prevalence of newly diagnosed CKD in the healthy population was analyzed, the causal relationship between sleep onset time and CKD could not be confirmed. Second, our study had strictly defined inclusion criteria; thus, the prevalence of CKD in participants was only about 5.5%, while that among Korean adults is approximately 8.2% [36]. Third, the information provided regarding sleep was based on a questionnaire rather than an objective measurement. Fourth, we did not analyze dialysis status and other sleep disorders, such as sleep apnea and restless leg syndrome. Fifth, the large difference in sample size between the two groups and the huge sample size can be important issues for the application and interpretation of statistics [37]. Since the P value converges to 0 as the sample size increases in the data, the confidence interval, clinical significance of the variable, and classical risk factors were taken into consideration when setting and interpreting variables, but these issues may still remain. Finally, due to the limitations on the diversity of survey items, the relationship between CKD and other lifestyles was not sufficiently elucidated (e.g., eating patterns, sitting time, working days per week, and working hours).

## 5. Conclusions

Nevertheless, this is the first large-scale cross-sectional study to examine the association between sleep onset time and CKD in >10,000 individuals who provided detailed information about their sleep parameters. The high incidence of new onset CKD in the inadequate bedtime groups strongly suggests that disturbances in circadian rhythm can contribute to CKD development. Further large-scale studies are needed to elucidate the precise role of a sleep schedule or the circadian system in CKD and to validate whether therapeutic strategies that modulate it are effective in CKD patients.

## Figures and Tables

**Figure 1 ijerph-19-05732-f001:**
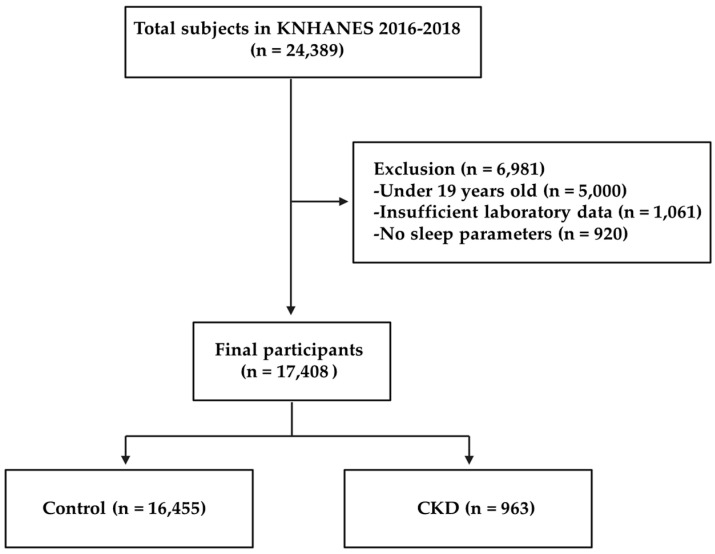
Flowchart of the study.

**Figure 2 ijerph-19-05732-f002:**
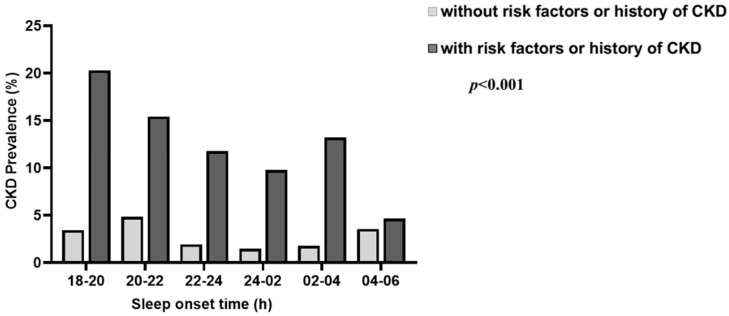
Distribution of CKD prevalence at different sleep onset times in participants with and without CKD risk factors or a history of CKD. Significance between each CKD incidence by sleep onset time was calculated by Pearson’s chi-square test in both groups with and without CKD risk factors or history.

**Figure 3 ijerph-19-05732-f003:**
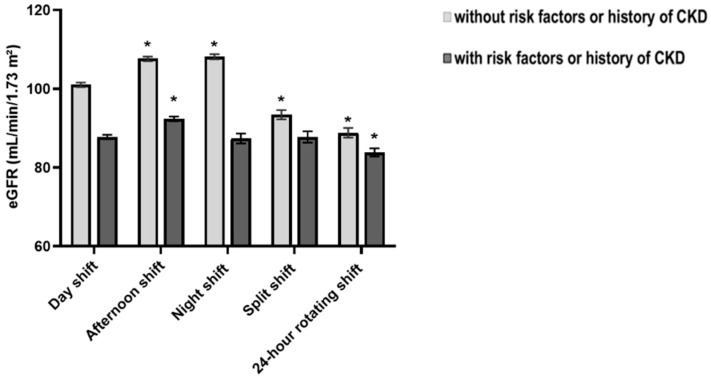
Estimated glomerular filtration rate (eGFR) in each work schedule. * *p* < 0.05 compared with day shift.

**Table 1 ijerph-19-05732-t001:** General characteristics of participants (*n* = 17,408).

	Control *(n* = 16,445)		CKD (*n* = 963)		*p*
	*n* (or MEAN)	% (or STD)	*n* (or MEAN)	% (or STD)	
SEX					<0.0001
Male	7078	43.04	515	53.48	
Female	9367	56.96	448	46.52	
Age	50.42	16.47	66.17	15.27	<0.0001
BMI	23.91	3.54	24.8	3.69	<0.0001
Education					<0.0001
Completion of middle school	4786	29.14	552	57.44	
Higher than high school	11,640	70.86	409	42.56	
Smoking status					<0.0001
No	9984	60.85	507	53.03	
Yes	6424	39.15	449	46.97	
Drinking status					<0.0001
No	1721	10.49	209	21.84	
Yes	14,692	89.51	748	78.16	
Physical activity					<0.0001
<2 days/week	15,076	91.69	935	97.19	
≥2 days/week	1366	8.31	27	2.81	
Diabetes					<0.0001
No	15,090	91.76	640	66.46	
Yes	1355	8.24	323	33.54	
Hypertension					<0.0001
No	12,789	77.77	349	36.24	
Yes	3656	22.23	614	63.76	
Sleep onset time					<0.0001
Early bedtime	1886	11.58	248	25.81	
Mid bedtime	13,044	80.11	660	68.68	
Late bedtime	1352	8.30	53	5.52	
Sleep duration					<0.0001
<6 h/day	2345	14.26	120	12.46	
6–<8 h/day	9169	55.76	475	49.33	
≥8 h/day	4931	29.98	368	38.21	
SBP	118.32	16.44	128.3	18.64	<0.0001
DBP	75.46	10.02	73.98	12.65	0.004
Cr	0.79	0.16	1.23	0.81	<0.0001
FBG	100.47	23.12	115.44	38.61	<0.0001
TG	134.53	110.24	159.96	126.96	<0.0001

The data are shown as *n* (%) for categorical variables or mean for continuous variables, and *p*-values were calculated using the chi-squared test or Student’s *t*-test. CKD, chronic kidney disease; *n*, number; STD, standard deviation; BMI, body mass index; SBP, systolic blood pressure; DBP, diastolic blood pressure; Cr, creatinine; FBG, fasting blood glucose; TG, triglyceride.

**Table 2 ijerph-19-05732-t002:** Odds ratio (OR) of the incidence of CKD according to sleep onset time and sleep duration.

	MODEL 1				MODEL 2				MODEL 3			
	OR	95% CI		*p*	OR	95% CI		*p*	OR	95% CI		*p*
Sleep onset time				<0.0001				0.016				0.046
Early bedtime	2.599	2.228	3.031		1.086	0.922	1.280		1.049	0.883	1.248	
Mid bedtime	REF				REF				REF			
Late bedtime	0.775	0.582	1.031		1.531	1.135	2.066		1.479	1.083	2.020	
Sleep duration				<0.0001				0.061				0.047
<6 h/day	0.988	0.804	1.213		0.887	0.718	1.095		0.852	0.685	1.058	
6 to <8 h/day	REF				REF				REF			
≥8 h/day	1.441	1.252	1.658		1.133	0.978	1.312		1.123	0.963	1.309	

Model 1: unadjusted; Model 2: adjusted by age, sex; Model 3: Model 2 + education level, smoking, drinking, physical activity, BMI, SBP, DBP, FBG, TG, and comorbidities (such as hypertension, diabetes, hyperlipidemia, cerebrovascular diseases, myocardial infarction, and angina pectoris); OR, odds ratio; CI, confidence interval; Ref, reference.

**Table 3 ijerph-19-05732-t003:** Newly diagnosed CKD prevalence among participants without CKD risk factors or a history of CKD according to sleep onset time.

	MODEL 1				MODEL 2				MODEL 4			
	OR	95% CI		*p*	OR	95% CI		*p*	OR	95% CI		*p*
Sleep onset time				<0.0001				0.005				0.037
Early bedtime	2.921	2.033	4.196		1.748	1.175	2.602		1.535	1.011	2.330	
Mid bedtime	REF				REF				REF			
Late bedtime	1.186	0.762	1.845		1.536	0.976	2.417		1.485	0.942	2.339	

Diabetes, hypertension, hyperlipidemia, and cardiovascular disease were identified as risk factors for CKD. Model 1: unadjusted; Model 2: adjusted by age and sex; Model 4: Model 2 + education level, smoking, drinking, physical activity, and BMI; OR, odds ratio; CI, confidence interval; Ref, reference.

## Data Availability

The Korea National Health and Nutrition Examination Survey (KNHANES) data are publicly available on the official KNHANES website (https://knhanes.kdca.go.kr, accessed on 19 November 2021).

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
