# Peer review of "Perturbation of Circadian Rhythm Is Associated with Increased Prevalence of Chronic Kidney Disease: Results of the Korean Nationwide Population-Based Survey"

_ijerph, 2022, doi:10.3390/ijerph19095732_

Round 1

Reviewer 1 Report

In introduction part the authors need to present in more detail the relationship between circadian rhythm and renal function and how cold be a link between that and development of CKD to understand better the aim of this study. Also, the authors must present primary and secondary endpoints of this study. In materials and methods part they did not specify how the terms drinking status and physical activity are defined.  On the statement in lines 158 and 159 the authors should explain how ckd appears in the absence of risk factors - what they refer to? The authors did not specify how many patients with CKD were on dialysis and whether there was a link between sleep time on set and dialysis status.

Author Response

Response to Reviewer 1 Comments

I, along with my coauthors, appreciate helpful comments and an opportunity to revise our manuscript. We have thoroughly reviewed your kind peer-review comments and have revised the article as follows.

Point 1: In introduction part the authors need to present in more detail the relationship between circadian rhythm and renal function and how cold be a link between that and development of CKD to understand better the aim of this study. Also, the authors must present primary and secondary endpoints of this study.

Response 1: Thank you for the advice, we have modified the introduction to further explain the relationship between circadian rhythm and renal function and its effects on CKD, and also clarified the purpose of the study. (Page 2, lines 62-85)

Point 2: In materials and methods part, they did not specify how the terms drinking status and physical activity are defined.  

Response 2: Thank you for indicating this point. We have added details of some parameters (drinking status, physical activity) to the manuscript. (Page 4, lines 142-145)

Point 3: On the statement in lines 158 and 159 the authors should explain how ckd appears in the absence of risk factors - what they refer to?

Response 3: Following your advice, the rationale of analyzing the prevalence of newly-onset CKD in a population without classical risk factors such as DM, HTN, and hyperlipidemia was further addressed. (Page 6, lines 201-210)

Point 4: The authors did not specify how many patients with CKD were on dialysis and whether there was a link between sleep time on set and dialysis status.

Response 4: We appreciate your comment. The relationship between sleep disturbances and dialysis can be an important issue in further looking at the effects of sleep disturbances on CKD. However, this analysis could not be performed because information on dialysis was not included in the survey. In this regard, we added this issue to the limitation. (Page 9, lines 331)

Reviewer 2 Report

The manuscript is well-written, concise, and straightforward.  I have only one major concern, and that is the application and interpretation of the statistics. Many of our commonly used statistical tests were not designed for use with enormous sample sizes and spurious findings can result.  For example, see here: https://www.jstor.org/stable/24700283

Given the disparity in sample sizes between CKD and control, and the huge sample sizes overall, it would be nice to have some sense of effect size / biological significance.  For example, is the difference in blood presssure biologically significant?

Author Response

Response to Reviewer 2 Comments

I, along with my coauthors, appreciate helpful comments and an opportunity to revise our manuscript. We have thoroughly reviewed your kind comments and have revised the article as follows.

The manuscript is well-written, concise, and straightforward. I have only one major concern, and that is the application and interpretation of the statistics. Many of our commonly used statistical tests were not designed for use with enormous sample sizes and spurious findings can result. For example, see here: https://www.jstor.org/stable/24700283. Given the disparity in sample sizes between CKD and control, and the huge sample sizes overall, it would be nice to have some sense of effect size / biological significance. For example, is the difference in blood pressure biologically significant?

Response: Thanks for your comment and introducing great reference. The large difference in sample size between the two groups and the huge sample size can be important issues for the application and interpretation of statistics. Since the P value converges to 0 as the sample size increases in the data, the confidence interval, clinical significance of the variable, and classical risk factors were taken into consideration when setting and interpreting variables, but these concerns may still remain. Following your advice, these concerns are additionally described in the discussion.

Reviewer 3 Report

Perturbation of Circadian Rhythm is Associated with Increased Prevalence of Chronic Kidney Disease: Results of the Korean Nationwide Population-Based Survey (ijerph-1664696)

This study aimed to evaluate the association between chronic kidney disease (CKD) and sleep disturbance in a Korean adult population. According to authors impaired circadian rhythm along with sleep disturbance can be associated with CKD development; therefore, sleep disturbance might be an important therapeutic target for CKD.

An interesting attempt to combine circadian rhythm disturbance and CKD. However, in the process of developing CKD, a series of molecular interactions, biochemical changes in the organism and comorbidities are observed. So, can the CKD development opportunity be related to the above factor? A very large cohort of participants was selected for the study, which was examined in many factors. I believe that not so much the quality of sleep as the lifestyle, for example, time of work and study, diet, can affect the work of the kidneys, and these factors were not included in the authors' study. also, there is a big difference between the number of people in the control group and the CKD patients.

Authors: "Participants in the CKD group were older and less educated and had a higher prevalence of diabetes and hypertension, higher BMI, and higher triglyceride levels. They were also less physically active, were current smokers, and consumed less alcohol". I believe that education does not influence the development of a disease, but rather awareness. Moreover, the last sentence shows that lower alcohol consumption may affect the development of CKD.

Table 1: the study is badly planned - there is a very high statistical significance in the age of the respondents, which may generate large changes in the organism.

Authors should take into account the variables I have provided and send the corrected article once again.

Author Response

Response to Reviewer 3 Comments

I would like to extend my deep appreciation to you for review. I thankfully found this round of editing a good chance of improving the paper.

Point 1: An interesting attempt to combine circadian rhythm disturbance and CKD. However, in the process of developing CKD, a series of molecular interactions, biochemical changes in the organism and comorbidities are observed. So, can the CKD development opportunity be related to the above factor?

Response 1: We totally agree with your comments. Because age, smoking, metabolic burden, and many comorbid conditions are involved in the pathogenesis of CKD, it is very important to consider the interrelationships between such variables. Therefore, by including age, smoking, BMI and risk factors for CKD such as diabetes, hypertension, hyperlipidemia, angina pectoris, myocardial infarction, and stroke in the multivariate logistic regression analysis (Model 3), the independent association between sleep onset and CKD was investigated. We apologize for confusing readers by not giving a detailed explanation of the analysis with comorbidities added. Therefore, we revised the discussion part. (Page 8, lines 270-274)

Point 2: A very large cohort of participants was selected for the study, which was examined in many factors. I believe that not so much the quality of sleep as the lifestyle, for example, time of work and study, diet, can affect the work of the kidneys, and these factors were not included in the authors' study. Also, there is a big difference between the number of people in the control group and the CKD patients.

Response 2: We think this is a novel topic. Unfortunately, however, the KNHANES did not survey these lifestyle factors such as eating habits, sitting time, and study time. We really appreciate your comments. However, we believe that further studies should be conducted on the effects of various lifestyles related to circadian disorders in the future. In addition, as you pointed out, the large difference in sample size between the two groups or the huge sample size can be important issues for the application and interpretation of statistics. Since the P value converges to 0 as the sample size increases in the data, the confidence interval, clinical significance of the variable, and classical risk factors were taken into consideration when setting and interpreting variables in the study, but these concerns may still remain. All issues mentioned above were further addressed in the discussion.

Point 3: Authors: "Participants in the CKD group were older and less educated and had a higher prevalence of diabetes and hypertension, higher BMI, and higher triglyceride levels. They were also less physically active, were current smokers, and consumed less alcohol". I believe that education does not influence the development of a disease, but rather awareness. Moreover, the last sentence shows that lower alcohol consumption may affect the development of CKD.

Response 3: We agree with your opinion. The link between CKD and lower education needs more discussion. To date, few studies have investigated the level of education and CKD, but the most recent cohort data of 6,078 participants observed that low education may accelerate eGFR reduction and increase the incidence of CKD (doi: 10.1093/ndt/gfy361). The study suggested that some risk factors smoking, poor diet, BMI, WHR, and high blood pressure could underlie this association. Although our study could not account for a causal relationship between them, based on your comments, we've added discussion to our revised manuscript. (Page 4, lines 156-159)

Regarding alcohol consumption, some studies have shown that low to moderate alcohol consumption can reduce the risk of CKD (doi: 10.1053/j.jrn.2019.01.011). But since our data could not quantify the amount of drinking in detail, caution is needed in interpreting the results, and both the potential benefits and harms of drinking should be reconsider. Therefore, in order to avoid confusion due to non-quantified data, the explanation of drinking was removed from the results.

Point 4: Table 1: the study is badly planned - there is a very high statistical significance in the age of the respondents, which may generate large changes in the organism.

Response 4: We agree with your comments. There was a significant difference between the two groups on age, which may affect statistical differences in several variables. We think that this is an important point. In order to alleviate these concerns about the effects of aging, when dividing CKD and control, CKD was redefined by applying the age-specific threshold for GFR suggested by a study group (doi: 10.1681/ASN.2019030238), and the difference between the two groups was reanalyzed. As a result of the analysis, similar statistical significance was observed in several variables, although the age difference decreased. We added this to Supplementary Table S1 and modified the results. (Page 4, lines 159-164)

Round 2

Reviewer 1 Report

It could be accepted in the present form.

Author Response

Thank you very much for your assessments.

Reviewer 3 Report

Thanks to the authors for their responses to my comments. I am satisfied with the explanations.

Author Response

(The authors gave the same response as above.)
